# The Use of Synthetic Graft for MPFL Reconstruction Surgery: A Systematic Review of the Clinical Outcomes

**DOI:** 10.3390/medsci11040075

**Published:** 2023-11-28

**Authors:** Panayiotis Tanos, Chryssa Neo, Edwin Tong, Andrea Volpin

**Affiliations:** 1Institute of Applied Health Sciences, University of Aberdeen, Aberdeen AB24 3FX, UK; p.tanos@outlook.com; 2Queen Elizabeth Hospital Gateshead, Gateshead Health NHS Foundation Trust, Gateshead NE9 6SX, UK; neochryssa@gmail.com; 3Aberdeen Royal Infirmary, NHS Grampian, Aberdeen AB25 2ZN, UK; edwin.tong@nhs.scot; 4Trauma and Orthopaedics, Dr Gray’s Hospital, NHS Grampian, Elgin IV30 1SN, UK

**Keywords:** acute patella dislocation, medial patellofemoral ligament, synthetic graft, MPFL reconstruction

## Abstract

(1) Background: Acute patella dislocation (APD) is a prevalent knee injury, with rates between 5.8–77.8 per 100,000. APD often results in repeat lateral patella dislocations due to the instability of the medial patellofemoral ligament (MPFL). Non-operative treatments have a 50% recurrence rate. While autologous grafting for MPFL has been favored, surgeons are now exploring synthetic grafts. We aimed to assess the effectiveness of synthetic grafts in MPFL reconstruction surgeries for repeated patellar dislocations; (2) Methods: Our research was based on a thorough search from the National Institute of Health and Clinical Excellence Healthcare Databases, using the Modified Coleman Methodology Score for quality assessment; (3) Results: Six studies met the inclusion criteria. A total of 284 patients and 230 knees were included. Seventy-five percent of patients were graded to have excellent-good clinical outcomes using the Crosby and Insall Grading System. International Knee Documentation Committee score and Knee injury and Osteoarthritis Outcome Score scores showed 59% and 60% post-operative improvement, respectively; (4) Conclusions: All studies showed improvement in post-operative functional outcomes and report no serious adverse events. The 6 mm, LARS (Orthomedic Ltd., Dollard-des-Ormeaux, QC, Canada) proved to have the most improvement in post-operative outcomes when used as a double bundle graft.

## 1. Introduction

Acute patella dislocation is described as the displacement of patella from the trochlear groove of the femur, often involving lateral displacement [1]. It is considered as one of the most common knee injuries with a prevalence of 5.8 to 77.8 per 100,000, and with highest incident rate in active adolescents [2,3,4]. These can lead to recurrent dislocations, patellar instability, persistent knee pain, decreased level of sport activities, and patellofemoral arthritis. Traditionally, the initial management for first-time patella dislocation is non-operative treatment, however, the rate of recurrence is estimated to be 50% [5,6].

The primary soft tissue stabilizer to lateral patella dislocation is the medial patellofemoral ligament (MPFL), first described in the late 1950s [7,8,9]. MPFL injury is complicated in over 90% of first-time patellar dislocations [10,11]. MPFL therefore exerts a more prominent role in recurrent instability than other predisposing anatomical morphological properties. Increasing developments in the use of synthetic grafts in MPFL have resulted in a rise of MPFL reconstruction surgery associated with satisfactory clinical outcomes [11,12].

Numerous surgical techniques using various types of grafts have been described in the literature to treat patella dislocations. Nevertheless, no single surgical technique has shown superiority in the literature. Regarding the types of graft, options are autologous grafting with either donor sites from gracilis, semitendinosus, TFL, or quadriceps tendons [13]. These have shown to produce positive outcomes with re-dislocation rates of 1.2% to 2.44% [14]. A systematic review of MPFL reconstruction by Shah et al. demonstrated 26.1% of complications to be haematoma, donor site infection, pain, and patella fractures [15]. One plausible solution to this issue may prevail in the use of synthetic graft for MPFL reconstruction in place of an autograft tendon.

As MPFL reconstruction gains further recognition, the operation with the most favorable results and minimum complications needs to be determined. The main objective of this review is to assess the clinical outcomes of patients with patella dislocations who had undergone MPFL reconstruction surgery with the use of synthetic graft. The primary outcome measure is the post-operative recurrence rate of dislocation. Secondary outcomes measures are the Kujala scores and post-operative complication rate. The secondary objective is to evaluate the difference between the available synthetic grafts in MPFL reconstruction surgery for recurrent patellar dislocations.

## 2. Materials and Methods

### 2.1. Inclusion Criteria

The articles were screened and included in our review if they met the following eligibility criteria: (1) Using the Oxford Center of Evidence-Based Medicine guidelines, studies with level of evidence I to IV were included; (2) Subjects were patients who had recurrent patellofemoral instability; (3) Subjects were patients who had undergone MPFL reconstruction using a synthetic graft; (4) Reporting of clinical and functional outcomes with reliable tools; (5) Minimum of 24-month follow up; (6) English language; (7) No minimum subject number in the studies

### 2.2. Exclusion Criteria

All animal or in vitro studies were excluded. Studies were excluded if it was revision surgery, an additional stabilization operation (ACL, PCL, MCL, LCL) at the time of MPFL reconstruction, if it focused on MPFL repair, and/or if it had patients with congenital disease or technical notes. Articles deemed as not presenting the full data were also excluded. Patients who received isolated MPFL reconstruction despite a severe anatomic risk factor constellation such as severe trochlear dysplasia and patella alta were excluded. Patients who received bony corrections in addition to MPFL reconstruction were also excluded.

### 2.3. Search Strategy

This review was performed in accordance with the Preferred Reporting Items for Systematic Reviews and Meta-Analyses (PRISMA) flowchart, as shown in Figure 1. A comprehensive search strategy was conducted using the National Institute of Health and Clinical Excellence (NICE) Healthcare Databases Advanced Search of four databases between 1 January 2000 and 1 September 2023 (PubMed, MEDLINE, Embase, CINAHL). The combination of subject headings and keywords carried out in the search strategy are (Patella OR Kneecap AND dislocation OR dislocations OR instability OR subluxation) AND (MPFL OR Medial Patellofemoral Ligament) AND (reconstruction OR ligament reconstruction) AND (Synthetic graft OR Synthetic material OR Synthetic Ligament OR Bioactive Synthetic Ligament OR Artificial Ligament OR LARS OR Polytape OR FiberTape).

### 2.4. Review Process

This yielded 337 results across four databases. These results were cross referenced and reduced to 260 after duplicates were deleted. Two authors independently screened the titles and abstracts. The full manuscripts were obtained for papers that were not excluded in this abstract reviewing phase. Any discrepancies or disagreements regarding the study inclusion were resolved by discussion. If consensus was not found, a third senior reviewer was available for a decision. Following this rigorous process, five studies were selected for inclusion in this study. The references of these full texts were assessed by the authors, identifying any other studies relevant to this topic.

### 2.5. Methodological Quality Assessment

The quality assessment for each of these observational studies were independently reviewed and scored by the two authors using the Modified Coleman Methodology Score (CMS). Part A of the CMS assesses the design of the study and Part B analyses the patient selection and outcome criteria. These 10 criteria give a total score between 0 and 100, where 100 indicates that the study largely avoids biases, chance, and confounding factors. It is further divided into excellent (85–100), good (70–84), fair (50–69), and poor (<50). Any discrepancy between the reviewers as to the scores for quality assessment were resolved by consensus of the three authors, including senior author AV.

## 3. Results

### 3.1. Study Characteristics

Seven studies met the inclusion criteria. The timeframe of the two papers published by E Nomura et al. had overlapped, thus, the paper with the higher number of knees was included to avoid duplication of patients. Two more studies were identified after editorial review. This brought the final number of studies to eight. All eight were single-center single-surgeon cohort studies, five of which were retrospective studies and three of which were prospective studies. The duration of follow up varied in the studies, from 24 months to 5.9 years. The earliest case reported was in 1991 and the most recent in 2023. The least follow up was 5.9 years, and the maximum up to 11.9 years (Table 1).

### 3.2. Patient Characteristics

A total of 327 patients were enrolled and 352 knees were operated on (Figure 1). This includes 203 female patients and 105 male patients, excluding 19 patients of unreported gender and 3 patients lost in follow-up. The average patient age was 24.7 years (9 to 44 years). Inclusion criteria for two studies were failure of three-month conservative management of quadricep muscle strengthening exercises. The other three studies included patients who had more than one episode of patellar dislocation. Exclusion criteria were evident osteoarthritic changes on radiographs and previous surgery to the same knee. All patients had MPFL reconstruction without any other ligamentous procedures performed simultaneously. Two studies included lateral release and one study included concomitant medial/distal anterior tibial tuberosity (ATT) transfer.

### 3.3. Methodological Quality Assessment

There were several methodological strengths and deficits that were identified in the studies reviewed. The mean Modified Coleman Methodological Quality Score (CMS) was 65.4, with a median score of 65 (overall range 57–71). Prospective studies were judged to be of higher methodological quality with higher recruitment rates. The primary reasons for fair scores in Part A of this assessment were study size, mean follow-up period, type of study, and description of diagnosis with percentage specified. In part B, the limitation was that the general health outcomes measures were rarely incorporated.

### 3.4. Intervention Characteristics

All studies used artificial ligaments in MPFL reconstruction from different manufacturers. This includes 15-mm wide tape-type Leeds-Keio artificial ligament, 20 mm × 500 mm Poly Tape (PT20; Neoligaments, Leeds, UK), second generation LARS R6 X 400 graft, FiberTape (Arthrex, Naples, FL, USA), and Mersilene Tape or AchilloCord Ligament (Table 2). In each of these papers, the procedure for MPFL reconstruction was performed by a single surgeon in a single center. Five studies employed a double bundle reconstruction, and three studies employed a single-bundle reconstruction.

Three studies employed a two-step surgical procedure (diagnostic knee arthroscopy, followed by reconstruction of the MPFL with a synthetic ligament). The other two studies had an open procedure undertaken for MPFL reconstruction. The majority used the through tunnel technique to fixate the artificial ligament, with two studies proposing a minimally invasive technique. Endobuttons were commonly applied to secure the surgical fixation; however, in one study, the double-stapling method was used instead. Our largest cohort study successfully placed the mid-section of the graft into the medial patella trough and used two 2.4-mm anchor sutures to secure the graft into the trough [21]. Another study compared the reduction technique of patella to the central or lateral of the trochlear during the MPFL reconstruction, which yielded similar results.

All but one study had a thorough description of the post-operative rehabilitation protocol. Not all studies specified if a knee immobiliser was used, but they included a period of non-weightbearing progressing to partial weightbearing before the patient can fully weight bear. Post-operatively, on day one, quadriceps rehabilitation exercises were started as tolerated by patient, progressing to passive knee range of motion. Light exercises and non-contact sports activities were permitted at week 8, and full sports activities from week 12 onwards.

### 3.5. Clinical Outcome Measures

Eight different methods were used to evaluate clinical outcomes post-operatively (Table 3). Most studies (4/8) used Kujala Anterior Knee Pain Score, where up to thirty percent improvement was demonstrated linearly in all three studies. Crosby and Insall Grading System was also commonly used. Most patients were graded to have excellent to good clinical outcomes (75%). Twelve percent were deemed to have excellent and nine percent good results post-operatively. Only four percent of patients were deemed to have fair outcomes and no patient was graded to have worse post-operative outcomes.

Lysholm Knee and Tegner Activity Scale were the second most popular outcome measure scales. The Lysholm Knee Scale demonstrated a 40% improvement post-operatively, whereas Tegner Activity Scale portrayed a three-point rise in activity levels; however, these results were not reproducible by the other studies. Finally, International Knee Documentation Committee (IKDC) score and Knee injury and Osteoarthritis Outcome Score (KOOS) scores showed a 59 and 60 percent significant post-operative improvement, respectively. Forty percent improvement was observed in the Social Functioning (SF-12) score. On the other hand, VAS score demonstrated a rapidly decreasing trend in mean score.

Migliorini et al. analyzed the role of synthetic graft in MPFL reconstruction by focusing on the Kujala, Lysholm, Tegner, and IKDC clinical scores as well as the rate of complications of the procedure in patients with recurrent patellofemoral instability. Significant post operative improvement was evident in all four clinical scores as well as minimally clinically important difference (MCID) at final follow-up. A low rate of complications was identified, of which positive apprehension test and persistent subjective sensation of instability were the most common. Rates of revision (1%) and re-dislocation (2.5%) were the most common and agree with our results. We further expanded on the work of Migliorini et al. by analyzing the impact of the different types of synthetic grafts as well as the operative technique. Reducing the patella to the strict center versus slightly laterally showed no significant difference; however, a minimally invasive technique portrayed several benefits over open reconstruction [25].

Milinkovic et al., 2023 reported an increase from 35.0  ±  21.7 to 79.7  ±  13.3 (*p*  <  0.0001) in the use of synthetic graft from preoperatively to postoperatively, without any significant difference from the pedicled quadriceps tendon autograft group [22].

### 3.6. Type of Synthetic Graft

Understandably it is cost efficient and environmentally friendly to use grafts produced in the country of surgical operation. However, sizes and materials of production differ and it is therefore essential to distinguish which type of synthetic ligament provides the best outcomes (Table 4). Ligaments vary in size (mm) and material. The decision on which type of ligament to use depends on availability, experience, and preference, but also on anatomical variation and extent of damage. When using a double graft bundle, the 6 mm, LARS (Orthomedic Ltd., Dollard-des-Ormeaux, QC, Canada) proved to be 17% more effective than the 20 mm, polyester, Poly-Tape PT20 (Neoligaments Ltd., Leeds, UK) and Ultra-high molecular weight polyester tape, FiberTape (Arthrex, Naples, FL, USA) using the Kujala Anterior Knee Pain Score. The 6 mm, LARS further proved to improve post-operative outcomes by 50% using the Tegner Activity Scale in comparison to the 20 mm, polyester, Poly-Tape PT20, which offered no change to post-operative outcomes. Single graft bundle LARS (CORIN Ltd., Montbonnot-Saint-Martin, France), AchilloCordPLUS (Neoligaments Ltd., Leeds, UK) showed excellent to good clinical outcomes in more patients and operations than the 15 mm, polyester, Leeds-Keio (Neoligaments Ltd., Leeds, UK), which had one fair outcome.

### 3.7. Double Bundle vs. Single Bundle

One hundred and fifteen single and 237 double graft bundles were used across all eight studies. There was a 4.3 times post-operative improvement in Lysholm Knee Scale in comparison to 1.3 times difference favoring single graft bundle. However, the study of Deo et al., 2023 demonstrated excellent clinical outcomes using double bundle grafts. Likewise, there were more excellent to good and good results in the Crosby and Insall Grading System for single graft bundles versus double graft bundles (Table 4) [21].

### 3.8. Complications

All studies referred to surgical complications or adverse events. There was only one report of patella re-dislocation 9 months following trauma in this sets of studies. Eleven patients experienced tenderness at the staple fixation. One patient experienced infrapatellar paraesthesia that was resolved completely in 2 months. Four patients had prominence of ligament at the medial femoral condyle. Three patients complained of anterior knee pain at 24 month follow up. One patient experienced persistent pain that was determined to stem from calcific formation at the femoral insertion site of MPFL, portrayed by the 8-month post-operative CT scan. Symptoms completely resolved following surgical debridement. All other patients with post-operative pain experienced relief following the removal of the screw anchor.

## 4. Discussion

For more than 20 years, synthetic scaffolds have been developed for tendon and ligament repair surgery. Ellera Gomes pioneered MPFL reconstruction in 1992 by using a synthetic graft for his procedure, starting his series with a Leeds Keio (LK) (Neoligaments, Leeds, UK) ligament, then replaced by an Artrolig (Engimplan-Engenharia De Implante E Com, Brazil) 8 mm tubular polyester graft [12]. To date, this is the most comprehensive review assessing the use of synthetic grafts in MPFL reconstruction surgery for recurrent patellar dislocations [26]. The main finding of this systematic review is that all studies showed improvement in functional outcomes in their cohort of patients. No serious adverse events were reported in any of the studies.

### 4.1. Double Bundle vs. Single Bundle

Recently, focus has shifted from pure anatomical reconstruction to the importance of establishing biomechanical function of ligaments. This is demonstrated in single versus double bundle reconstructions. The combination of the superior-oblique bundle and the vastus medialis obliquus allows for the maintenance of the dynamic patellar stability. Simultaneously, the inferior straight bundle gives static strength of inhibition. This angular synergy effect allows for greater resistance to dislocation [24]. Wang et al. demonstrated the importance of double bundle reconstruction using both the Kujala score and the subjective questionnaire score. They further comment that recurrence of patellar dislocation was only observed in single bundle reconstructions and instability was 6.5 times higher in the single bundle reconstructions [27]. However, so far in the literature there is no evidence that shows MPFL reconstruction with a quadriceps tendon strip, which is equivalent to single-bundle technique, to be inferior to double-bundle techniques with a semitendinosus or gracilis graft. Interestingly, Kita et al. note that severe trochlear dysplasia and tibial tubercle–trochlear groove (TT-TG) distance are two factors that could further affect double bundle reconstruction [28]. Our cohort was not deemed sufficient to make a clear comparison between single and double bundle reconstructions. Berruto et al. used both techniques according to the severity of subluxation [16]. In the current literature, the ideal biomechanical properties (stiffness, viscoelasticity, tensile strength, thickness) of a graft for MPFL reconstruction remains largely undefined. Factors such as the patient’s pathoanatomical risk factors and bone morphology must be considered in selecting the optimal type and size of the synthetic graft as well as the overall surgical plan. Despite not extensively being analyzed in this systematic review, it is worth mentioning that in cases of trochlear dysplasia, trocheoplasty may be considered to deepen the trochlear groove, that in patella alta, distalizing tibial tubercle osteotomy may be performed to normalize patellar height, and that in increased TT-CG distance, a medializing tibial tubercle osteotomy can be considered to correct this distance and realign the extensor mechanism.

In comparison between synthetic graft and autologous graft, the study by Lee et al. showed improvements across all scoring modalities between pre- and post-operative periods. By using synthetic grafts, tendon harvesting would not be required, and this eliminates potential complications related to donor site. Furthermore, autologous grafts are collagenous in nature and would likely undergo stretching over time, whereas synthetic grafts, such as Fibertape (FT), have predictable biomechanical properties. This information makes it crucial to avoid overtightening of the synthetic graft during graft tensioning. This is also very important for MPFL reconstruction with an autologous tendon graft. Lee et al. suggest tensioning the MPFL graft under direct arthroscopic vision to observe the patella position over the trochlea without the use of a thigh tourniquet [17].

### 4.2. Surgical Technique

Apart from the comparison of autologous graft against synthetic graft, the differences in surgical technique may also play a part in the effectiveness of MPFL reconstruction. Suganuma et al. [18] compared between patella that were reduced to the strict center and those which were slightly lateral to the center of the trochlear. These comparisons were both conducted using synthetic graft in the reconstruction of MPFL. There was no significant difference in knee function scores between them, although better subjective evaluations existed in knee joints where patella fixed slightly lateral.

Three surgeons opted to use an open procedure for their MPFL reconstruction [16,18,19]. However, two authors employed a minimally invasive technique [20,21]. With minimal incisions, one surgeon was able to avoid violating the extensor mechanism, as compared to an open surgical technique [20]. Furthermore, a minimally invasive technique helps to reduce post-operative swelling, reduced pain, reduced risk of complications, earlier recovery times, and the ability to undergo post-operative rehabilitation without a knee immobilizer. Hersh et al. comments that the trough used encourages tissue ingrowth, which ensures the attachment of the graft and sutures at the medial patella border [21].

### 4.3. Limitations

A limitation of this systematic review includes the low sample size, which does not allow for a power calculation. Still, this is the biggest sample collected on synthetic MPFL reconstruction, giving a very concise picture of the results in literature. A further limitation is the lack of control and direct comparison between autologous, conservative, and other reconstruction techniques. Unfortunately, this will be hard to establish in the long term, since as technology develops, the standard of treatment improves with it and patients will all naturally start receiving the best possible treatment with the least side effects. Another limitation that we faced would be the lack of long-term follow-up. The average duration of follow up between these studies would be 50.1 months. Given the positive outcomes of MPFL reconstruction, achieving a long-term follow up for patients will be difficult. Having a mid-term study as such may inevitably result in estimates that are less reliable or precise. Furthermore, all the studies included in this review are single-centred and performed by a single surgeon. As a result of this, we were unable to compare between the use of autologous grafts against allografts as well as different type of surgical techniques used.

## 5. Conclusions

At present, these clinical studies support the use of artificial ligaments in MPFL reconstruction for recurrent patellar dislocations due to their optimistic short to medium term clinical outcomes, safety profile and reduction in risks of graft site complications. Nevertheless, there were insufficient high-quality studies and the small sample sizes could also account for the inaccuracies in the results. This study highlights the importance for further well-planned, long term, multicenter, prospective RCTs to be conducted, so more evidence can be collated to support the superiority of artificial ligaments over autografts.

## Figures and Tables

**Figure 1 medsci-11-00075-f001:**
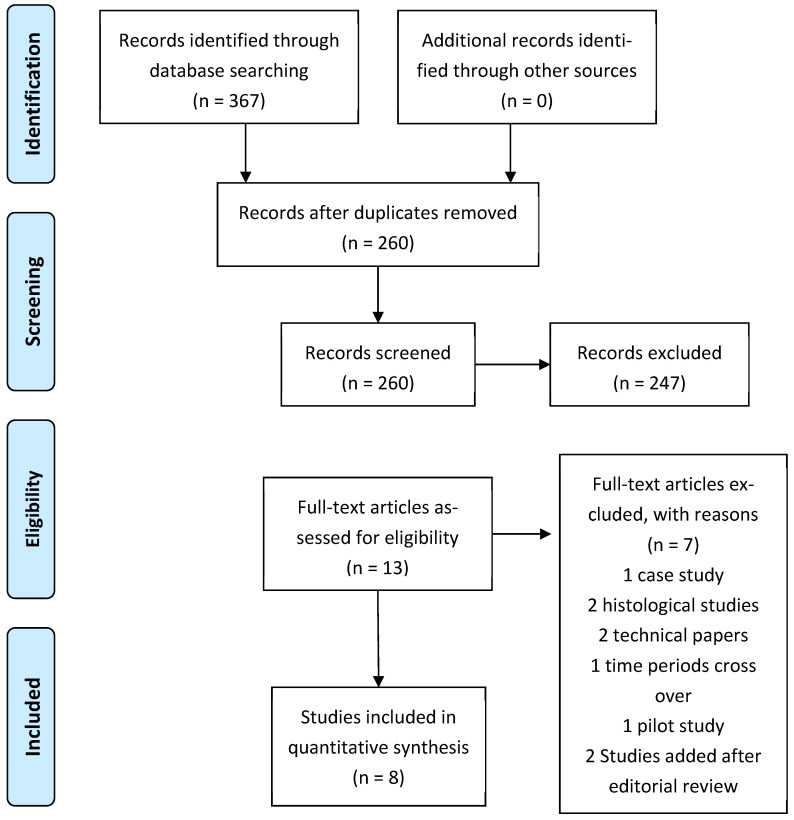
Preferred Reporting Items for Systematic Reviews and Meta-Analyses (PRISMA) flowchart.

**Table 1 medsci-11-00075-t001:** Study and Patient Characteristics; patella dislocation (PD), conservative treatment (CT), imaging findings of torn MPFL or trochlear dysplasia (IF), High grade Trochlear Dysplasia (TD), Associated menisci tears (AMT) ACL deficiency (ACL-D), Previous surgery to ipsilateral knee (PS), Osteoarthritis (OA), use of muscle relaxants (MR), previous tibial tubercle osteotomy and/or other bony procedures at the distal femur or proximal tibia (TTO), previous knee ligament surgical procedures (PLS) Trochlea dysplasia (TD), Patella lta (PA), lateral patellar instability (LPI).

Reference, Year	Study Design (Level of Evidence)	No. of Patients (No. of Knees)	Female/Male	Mean Age (Years)	Inclusion	Exclusion	Follow Up	CMS
Berruto et al., 2014 [16]	Prospective case series (IV)	16 (18)	7/9	19.0	>1 × PD	ACL-D, AMT	40.6 months	70
Lee et al., 2017 [17]	Prospective cohort study (II)	22 (23)	15/7	21.0	>1 × PD	PS	48 months	65
Suganuma et al., 2016 [18]	Retrospective case series (IV)	46 (46)	30/16	20.5	3–5 × PD	OA, MR, PS	48 months	64
Nomura et al., 2000 [19]	Prospective case series (IV)	24 (27)	19/5	21.0	Failed CT > 3 months	OA	5.9 years	71
Khemka et al., 2016 [20]	Retrospective case series (IV)	29 (31)	11/18	25.0	Failed CT > 3 months IF	PS, OA	43 months	57
Deo et al., 2023 [21]	Retrospective cohort study (IV)	85 (85)	27:58	28.0	>2 × PD Failed CT IF	PA, TD Malalignment	12–108 months	62
Milinkovic et al., 2022 [22]	Retrospective cohort study (III)	57 (57)	18/39	26.0	≥2 PD Failed CT > 6 months IF	PS, TTO, PLS.	2 years	70
Sasaki et al., 2022 [23]	Retrospective cohort study (IV)	43 (65)	14/31	N/A	LPI +/− PD	TD	1 year	57

**Table 2 medsci-11-00075-t002:** Intervention Characteristics, Full sports (FS), passive and active range of motion (P/AROM), partial and full weight nearing (P/FWB), Immediate knee immobilizer (IKI), quad exercises (QE), Patellar brace (PB), Progressive rehab (PR), medial epicondylar (ME), superficial infection (SI), residual instability (RI), medial knee pain (MKP), kyloid scar (KS), subcutaneous haemarthrosis (SH), localised tenderness at staple fixation (LT), infrapatellar paraesthesia (IPP), Anterior knee pain (AKP), medial femoral condyle ligament prominence(LP), patellar subluxation (PS).

Reference, Year	Graft Fixation	Operative Technique	Post-Op Rehab Protocol	Complications	Management
Berruto et al., 2014 [16]	7 mm Biorci re-sorbable interference screw	Open procedure—MPFL reconstruction +/− lateral release +/− medial +/− distal ATT transfer, performed with lateral release 2× patella tunnels, 20 mm in between	PWB Immediate PROM day 0 FWB week 6 PR for 6 months	1 MKP 7 Discomfort due to calcific formation at femoral insertion of MPFL	1× revision debridement 7× ATT screws removal
Lee et al., 2017 [17]	4.75 mm PEEK Swive-Lock bone anchor	EUA + diagnostic arthroscopy, then open. 2× patella tunnels. Transfemoral tunnel insertion point	Unspecified	1 KS. 2 SI	Conservative
Suganuma et al., 2016 [18]	Double staplers	EUA + diagnostic arthroscopy, then open	Knee brace PROM day 0 PWB + AROM day 2 FWB day 5 FS week 12.	1 IPP	Resolved in 2 months
Nomura et al., 2000 [19]	Double staplers	Open procedure—MPFL reconstruction +/− lateral release. A retinacular slip in distal vastus medialis. Double stapling method. Suction drain for operations with lateral release.	IKI QE day 1, PROM day 2. PB day 5, WB as tolerated FWB day 10. MS week 8 FS week 12.	11 LT 1 SH 1 SI	Conservative
Khemka et al., 2016 [20]	2× standard AO screws	Minimally invasive technique. EUA + diagnostic arthroscopy, small incisions, Through tunnel technique.	No IMI PWB and QE from day 1, PROM week 1. MS week 6 FS week 12.	1 trauma 4 LP 3 AKP	Patella re-dislocation for trauma managed surgically
Milinkovic et al., 2022 [22]	nonresorbable suture tape	Minimally invasive technique Passed through the proximal and distal origin of the native MPFL, two topstitching seams added to proximal and distal edges of the synthetic graft. Through tunnel technique	WB 3–4 weeks FWB 5–6 weeks	No significant complications reported	N/A
Sasaki et al., 2022 [23]	two 3.5-mm SwiveLock knotless anchors	diagnostic arthroscopy lateral retinacular release lateral retinacular release	ROM early QE PR FS 2 months	2 LT 3.2% PS	Conservative
Deo et al., 2023 [21]	Two 2.4-mm suture anchors	Minimally invasive technique—MPFL reconstruction. Through tunnel technique. Socket is made in the medial femur. The tape limbs are then double breasted and whip stitched 20 mm beyond Schottle’s point.	Unspecified	5 MKP 2 RI 2 SI	Conservative

**Table 3 medsci-11-00075-t003:** Clinical Outcome Measure.

Authors	Scores	Numerical Improvement Pre- to Post-Operatively	Conclusion
Suganuma et al., 2016 [24] Berruto et al., 2014 [16] Lee et al., 2017 [17]	Kujala Anterior Knee Pain Score	57 ± 8.4 (44–73) to 84.3 ± 10.2 (62–100) increase x1/3 post-operatively mean 64 (SD 14) to 84 (SD 18)	Significantly Improved *p* < 0.01
Lee et al., 2017 [17]	Tegner Activity Scale	median 3 to 6	Improved
Suganuma et al., 2016 [18]		no change	
Lee et al., 2017 [17]	Lysholm Knee Scale	61 (SD 15) to 80 (SD 9)	Improved
Khemka et al., 2016 [20]		20 (SD 19) to 87 (SD 9)	
Nomura et al., 2000 [19]	Crosby and Insall Grading System	50% excellent, 40% good, 10% fair	Improved
Khemka et al., 2016 [20]		96% excellent—good, 4% fair	
Berruto et al., 2014 [16]	IKDC Score	42.4 ± 7.1 (28.7–50.6) to 70.1 ± 3.9 (41.4–85.1)	Significantly Improved *p* < 0.01
Berruto et al., 2014 [16]	KOOS score	62.7 ± 4.3 (55.4–69) to 82.8 ± 8.8 (58.3–92.3)	Significantly Improved *p* < 0.01
Sasaki et al., 2022 [23]		54.7 ± 29.1 to 92.0 ± 12.9	
Milinkovic et al., 2022 [21]	Banff Patella Instability Instrument 2.0	35.0 ± 21.7 to 79.7 ± 13.3	Significantly Improved *p* < 0.01
Lee et al., 2017 [17]	SF-12	mean 48 (±15) to 66 (±10)	Improved
Berruto et al., 2014 [16]	VAS score	mean 2.5 ± 1.6 (0–8) to 1.4 ± 1.5 (0–6)	No Improvement
Deo et al., 2023 [21]	Kujala Oxford knee scores	mean 42.12 ± 12.55 to 78.79 ± 14.92 mean 23.15 ± 5.43 to 38.62 ± 6.68	Significantly Improved *p* < 0.01

**Table 4 medsci-11-00075-t004:** Comparison between types of synthetic grafts.

Reference, Year	Surgeon	Type of Ligament	Graft Bundle	Crosby and Insall Grading System	Tegner Activity Scale	Kujala Anterior Knee Pain Score	Lysholm Knee Scale
Berruto et al., 2014 [16]	Single surgeon	6 mm, LARS (Orthomedic Ltd., Dollard-des-Ormeaux, QC, Canada)	Double	9 excellent 7 good 2 fair	N/A	mean 57 ± 8.4 to 84.3 ± 10.2 (47%)	N/A
Lee et al., 2018 [17]	Single surgeon	Ultra-high molecular weight polyester tape, FiberTape (Arthrex, FL, USA)	Double	N/A	median 3 to 6 at 48 months	mean 64 (±14) to 84 (±18) at 48 months (30%)	61 (±15) to 80 (±9) at 24 months 78 (±12) at 48 months
Suganuma et al., 2016 [18]	Single surgeon	20 mm, polyester, Poly-Tape PT20 (Neoligaments Ltd., Leeds, UK)	Double	N/A	no change	increase x1/3 post operatively	N/A
Nomura et al., 2000 [19]	Single surgeon	15 mm, polyester, Leeds-Keio (Neoligaments Ltd., Leeds, UK)	Single	26/27 excellent to good 1/27 fair	N/A	N/A	N/A
Khemka et al., 2016 [20]	Single surgeon	LARS (CORIN Ltd., France), AchilloCordPLUS (Neoligaments Ltd., Leeds, UK)	Single	excellent to good	N/A	N/A	20 (±19) to 87 (±9)
Milinkovic et al., 2022 [22]	Single surgeon	Nonresorbable sutures (FiberTape^®^, Arthrex Co., Nepales, FL, USA)	Single	N/A	N/A	N/A	N/A
Sasaki et al., 2022 [23]	Single surgeon	polyester high-strength suture tape (FiberTape; Arthrex) with knotless anchors (SwiveLock; Arthrex)	Double	N/A	N/A	Improved in 1 year	N/A
Deo et al., 2023 [21]	Single surgeon	(Xiros, Leeds, UK)	Double	excellent clinical outcomes	N/A	mean 42.12 ± 12.55 to 78.79 ± 14.92	N/A

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
