# Peer review of "The Use of Synthetic Graft for MPFL Reconstruction Surgery: A Systematic Review of the Clinical Outcomes"

_medsci, 2023, doi:10.3390/medsci11040075_

Round 1
Reviewer 1 Report
Comments and Suggestions for Authors
Dear Authors
I have reviewed your paper with great interest.
I will accept your paper after a minimal revision.
My revision is:
Title: Very Good
Abstract: Very Good
Introduction and AIM: The problem and the aim are well descripting.
Results: Focus on and well described.
Discussion and Thread: effectiveness Focus ON.
The assessment of outcomes in the Knee Injury, patella dislocation or fracture have the worst outcomes.
References: Well chosen but to improve
Figures and Table: Very Good.
Author Response
We would like to thank Reviewer #1 for reviewing our paper and recognising the benefits of our work upon the existing literature.
References have been improved and new references have been added.
Reviewer 2 Report
Comments and Suggestions for Authors
Dear authors!
This is a very interesting review highlighting the use of synthetic ligaments in MPFL plasty.
The review itself is well designed and well written.
Accept.
Comments on the Quality of English LanguageEnglish quality is fine
Author Response
We would like to thank Reviewer #2 for reviewing our paper and recognising the benefits of our work upon the existing literature.
Reviewer 3 Report
Comments and Suggestions for Authors
Dear Editor and authors,
I received this manuscript for review with great interest and recommend approval with major revisions.
In their systematic review, the authors analyzed the use of synthetic grafts for MPFL reconstruction. They conclude that the use of synthetic grafts for MPFL reconstruction leads to a good functional outcome. MPFL reconstruction using synthetic grafts is a clinically relevant topic, accordingly, the submitted work is in my opinion relevant for the Med.Sci. readership. But the manuscript still has a few weaknesses that need to be corrected before publication:
A total of 6 studies were included in the systematic review, but in my view, at least two recent publications were forgotten:
1.) Milinkovic, D.D., Zimmermann, F. & Balcarek, P. Medial patellofemoral ligament reconstruction using nonresorbable sutures yields comparable outcomes to reconstruction with a pedicled quadriceps tendon autograft when performed in addition to bony risk factor correction. Knee Surg Sports Traumatol Arthrosc 31, 264–271 (2023). https://doi.org/10.1007/s00167-022-07104-1
2.) Sasaki E, Kimura Y, Sasaki S, Yamamoto Y, Tsuda E, Ishibashi Y. Clinical outcomes of medial patellofemoral ligament reconstruction using FiberTape and knotless SwiveLock anchors. Knee. 2022 Aug; 37:71-79. doi: 10.1016/j.knee.2022.05.011. Epub 2022 Jun 10. PMID: 35696836.
Some smaller details will be discussed in the following:
Introduction:
Lines 41-42: This statement is inaccurately formulated. In this regard, there are some data that mention MPFL injuries in more than 90% after the first patellar dislocation (Keppler et al., AJSM 2011; Elias et al., Radiology 2002).
Lines 46-49: Again, further literature supporting the correct statement is useful.
Line 50: The systematic reviews by Mackay ND et al, Orthop J Sports Med. 2014 and Schneider DK et al, Am J Sports Med. 2016 reported re-dislocation rates of 1.2% to 2.44%.
Lines 51-52: This is not comprehensible. There is a risk of infection with any surgical procedure. No matter whether it is a tendon graft or a synthetic graft.
Materials and Methods:
Lines 74-78: Were patients who received isolated MPFL reconstruction despite a severe anatomic risk factor constellation (e.g., severe trochlear dysplasia, patella alta, etc.) excluded? Were patients who received bony corrections in addition to MPFL reconstruction excluded? If these patients were not excluded, it should be explained why this was not done.
Line 89: It is “FiberTape“ not “FibreTape”.
Results:
Line 189: Table 3 is not exactly clearly structured. It would make sense to present the results more briefly. But also the tables 1+2 should be designed more clearly.
Discussion:
Lines 251-254: Although the findings of Wang et al. are impressive, it should be mentioned that there is enough research to show that MPFL reconstruction with a quadriceps tendon strip (equivalent to single-bundle technique) is not inferior to double-bundle techniques with a semitendinosus or gracilis graft.
Lines 263-264: To what extent do the anatomical risk factors of patellar instability influence the choice of graft? Should they not be rather corrected if they are pronounced and clinically relevant?
Lines 270-271: It is absolutely correct that overtightening of MPFL reconstruction with a synthetic graft should be avoided under all circumstances. However, it should be mentioned here that this also counts for MPFL reconstruction with an autologous tendon graft. Technical errors, such as femoral tunnel malposition and overtightening are the most common reasons for complications after MPFL reconstruction with an autologous tendon graft (see Zimmermann et al., AJSM 2020; Feucht et al., OJSM 2020).
Line 276: Not “MPFL repair” but “MPFL reconstruction”.
Kind regards.
Author Response
Response: We would like to thank Reviewer #3 for reviewing our paper and recognising the benefits of our work upon the existing literature. We appreciate the extensive and analytical input which has helped us improve the quality of our article. We have now included the additional references which could potentially been omitted due to their very recent publication.
We have revised and reviewed all the following points to improve clarity and comprehension of our article.
Introduction:
Lines 41-42: This statement is inaccurately formulated. In this regard, there are some data that mention MPFL injuries in more than 90% after the first patellar dislocation (Keppler et al., AJSM 2011; Elias et al., Radiology 2002).
We have now amended as below:
‘’ MPFL injury is complicated in over 90% of first-time patellar dislocations’’
Lines 46-49: Again, further literature supporting the correct statement is useful.
We have now amended as below:
‘’ MPFL therefore exerts a more prominent role in recurrent instability than other predisposing anatomical morphological properties. Increasing developments in the use of synthetic grafts in MPFL have resulted in a rise of MPFL reconstruction surgery associated with satisfactory clinical outcomes’’
Line 50: The systematic reviews by Mackay ND et al, Orthop J Sports Med. 2014 and Schneider DK et al, Am J Sports Med. 2016 reported re-dislocation rates of 1.2% to 2.44%.
We have now amended as below:
‘’These have shown to produce positive outcomes with re-dislocation rates of 1.2% to 2.44%’’
Lines 51-52: This is not comprehensible. There is a risk of infection with any surgical procedure. No matter whether it is a tendon graft or a synthetic graft.
This statement has been removed to avoid confusion.
Materials and Methods:
Lines 74-78: Were patients who received isolated MPFL reconstruction despite a severe anatomic risk factor constellation (e.g., severe trochlear dysplasia, patella alta, etc.) excluded? Were patients who received bony corrections in addition to MPFL reconstruction excluded? If these patients were not excluded, it should be explained why this was not done.
We have now amended as below:
‘’Patients who received isolated MPFL reconstruction despite a severe anatomic risk factor constellation such as severe trochlear dysplasia and patella alta were excluded. Patients who received bony corrections in addition to MPFL reconstruction were also excluded.’’
Line 89: It is “FiberTape“ not “FibreTape”.
This has now been corrected
Results:
Line 189: Table 3 is not exactly clearly structured. It would make sense to present the results more briefly. But also the tables 1+2 should be designed more clearly.
All tables were extensively reviewed and revised to make them both more succinct and comprehensible
Discussion:
Lines 251-254: Although the findings of Wang et al. are impressive, it should be mentioned that there is enough research to show that MPFL reconstruction with a quadriceps tendon strip (equivalent to single-bundle technique) is not inferior to double-bundle techniques with a semitendinosus or gracilis graft.
We have now amended as below:
‘’ However, so far in literature there is no evidence of MPFL reconstruction with a quadriceps tendon strip which is equivalent to single-bundle technique to be inferior to double-bundle techniques with a semitendinosus or gracilis graft.’’
Lines 263-264: To what extent do the anatomical risk factors of patellar instability influence the choice of graft? Should they not be rather corrected if they are pronounced and clinically relevant?
We have now amended as below:
‘’ Factors such as the patient’s pathoanatomical risk factors and bone morphology must be considered in selecting the optimal type and size of the synthetic graft as well as the overall surgical plan. Despite not extensively being analysed in this systematic review its worth to mention that: in cases of trochlear dysplasia, trocheoplasty may be considered to deepen the trochlear groove; in patella alta, distalizing tibial tubercle osteotomy may be performed to normalize patellar height; and in increased TT-CG distance, a medializing tibial tubercle osteotomy can be considered to correct this distance and realign the extensor mechanism.‘’
Lines 270-271: It is absolutely correct that overtightening of MPFL reconstruction with a synthetic graft should be avoided under all circumstances. However, it should be mentioned here that this also counts for MPFL reconstruction with an autologous tendon graft. Technical errors, such as femoral tunnel malposition and overtightening are the most common reasons for complications after MPFL reconstruction with an autologous tendon graft (see Zimmermann et al., AJSM 2020; Feucht et al., OJSM 2020).
Following sentence was added:
‘’ This is also very important for MPFL reconstruction with an autologous tendon graft.’’
Line 276: Not “MPFL repair” but “MPFL reconstruction”.
This has been corrected